# Performance of mid-upper arm circumference as a screening tool for identifying adolescents with overweight and obesity

Binyam Girma Sisay[1], Demewoz Haile[1], Hamid Yimam Hassen[2], Seifu Hagos Gebreyesus[1]*

1 Department of Nutrition and Dietetics, School of Public Health, Addis Ababa University, Addis Ababa, Ethiopia, 2 Department of Primary and Interdisciplinary Care, Faculty of Medicine and Health Sciences, University of Antwerp, Antwerp, Belgium

* seif_h23@yahoo.com

**Data Availability Statement:** All relevant data are within the manuscript and its Supporting Information files.

## Abstract

### Background

Adolescent overweight and obesity is a global public health problem, associated with an increased risk of metabolic syndrome. Recently, mid-upper arm circumference (MUAC) has been suggested as a screening tool to identify overweight and obesity among school-age children and early adolescents (5–14 years). However, little is known about the potential use of MUAC in the late adolescence period (15–19 years). Therefore, the present study aimed to evaluate the performance of MUAC to identify overweight (including obesity) in the late adolescence period in Ethiopia.

### Methods

We conducted a cross-sectional study among 851 adolescents aged 15 to 19 years. We collected anthropometric data including MUAC, weight and height with the help of trained field workers. The receiver operating characteristic (ROC) curve analysis was used to examine the validity of MUAC compared to BMI Z score in identifying adolescents with overweight or obesity. Furthermore, we calculated the sensitivity, specificity, positive predictive value (PPV), negative predictive value (NPV), proportion of correctly classified, positive, and negative likelihood ratio for the proposed optimal cut-offs.

### Results

MUAC was strongly correlated with BMI Z score with a correlation coefficient (r) of 0.81 (95% CI; 0.79–0.84). The optimal MUAC cut-off for identifying adolescents with overweight or obesity was 27.7 cm for males and 27.9 cm for females. The area under the ROC curve (AUC) was 0.96 (95% CI; 0.93–0.98) for males and 0.96 (95% CI; 0.94–0.98) for females. The accuracy level of MUAC to identify adolescents with overweight (including obesity) was high for both sexes (overall a sensitivity of 91.1% and a specificity of 90.3%).

**Funding:** BGS have received financial support from Addis Ababa University for data collection. The funder had no role in study design, data collection and analysis, decision to publish, or preparation of the manuscript. There was no additional external funding received for this study.

**Competing interests:** The authors have declared that no competing interests exist.

## Conclusions

MUAC has relatively equivalent accuracy with BMI Z score to identify overweight and obesity in adolescents. Hence, MUAC could be used as an alternative tool for surveillance and screening of overweight in adolescents aged 15–19 years.

## Introduction

Adolescent overweight and obesity is a major public health problem with far-reaching and long-term adverse health outcomes [1]. Overweight and obesity are a disorders of positive energy balance commonly caused by the consumption of high-energy foods and sedentary behavior, combined with an inherent susceptibility to weight gain [2]. Adolescents with overweight and obesity are prone to be obese in their adulthood and are at higher risk of developing non-communicable diseases including high blood pressure, type 2 diabetes, cardiovascular diseases, sleep apnea, and cholelithiasis [3–5].

Body mass index (BMI) Z score is a widely used measure to identify overweight and obesity in school-age children and adolescents (5–19 years) [6]. Despite its popularity, BMI Z score is less preferred by minimally trained healthcare workers, and its measuring equipment are relatively expensive and require regular calibration. Moreover, it is time consuming to measure weight, height, and interpret the value with a reference chart [7, 8]. An important approach to promote early identification and surveillance of overweight and obesity among adolescents is developing an easy to use, inexpensive and reliable screening tool for identifying adolescents with overweight and obesity [9].

MUAC is a simple and cheap screening tool used to identify moderate and severe acute undernutrition among under-five children (6–59 months of age) in low and middle-income countries [10]. MUAC has also been used for numerous years as a screening tool for identifying undernutrition among under-five children and adults in situations, such as famines and refugee crisis, where height and weight measurements are difficult to perform [11]. Pregnant women's nutritional status, both undernutrition and obesity could reliably be assessed using MUAC in low-resource settings [12].

MUAC has the potential to be a practical, low cost, simple, and reliable measuring tool to identify adolescents with overweight and obesity. Few studies have indicated that MUAC is a valid measure to identify overweight and obesity in school-age children (5–9 years) and early adolescents (10–14 years) [8, 13, 14]. However, little is known about the ability of MUAC to identify overweight and obesity among adolescents aged 15–19 years. The present study aimed to evaluate the performance of MUAC to identify overweight in the late adolescence period (15–19 years) in Ethiopia.

## Materials and methods

### Study setting, design, and participants

A school-based, cross-sectional study was conducted among high school adolescents aged 15 to 19 years in selected public and private high schools of Addis Ababa. Addis Ababa is the capital city of Ethiopia; with a population of homogeneous racial identity. The city is divided into 10 sub-cites. There are 635,903 adolescents, of this 385,713 are between the age of 15–19 years [15]. The city has a total of 219 high schools, of which 73 are public and 146 are private schools.

Adolescents, aged 15–19 years, who were attending classes in the selected private and public high schools (Grade 9–12) of Addis Ababa were eligible to be included in the study. Whereas, adolescents with physical deformity that could affect height and weight measurement were excluded from the study. Besides, those who refused any of the anthropometric measurements were also excluded.

## Sample size and sampling procedure

The sample size was determined by using the diagnostic accuracy test study sample size calculation formula [16], assuming a sensitivity of 95.2%, a specificity of 89.9% [8], and a prevalence of overweight/obesity, 13.9% among adolescent students in Addis Ababa [17], with a 5% margin of error, a design effect of 1.5 and 10% non-respondent rate. Based on these assumptions, the final sample size was 877.

A multistage sampling technique was employed to select adolescents for the study. A total of 15 schools (10 private and 5 public schools) were selected using the lottery method, then the samples were distributed proportionally between public (546 participants) and private schools (327 participants). Four sections from each selected school (one from each grade level) were selected randomly. Finally, we used the students list to randomly select study participants from each section.

## Study procedures and measurements

Anthropometric measurements were performed by trained field workers, using standard techniques [18]. MUAC was measured on the non-dominant arm using non-stretchable plastic tape at the midpoint between the olecranon and the acromial process after the arm is flexed to 90 degrees from the elbow. Then, the arm was relaxed, the MUAC tape was placed around the marked midpoint of the arm, neither too loss nor too tight and the measurement was recorded to the nearest 0.1 cm. MUAC was measured twice for each subject and the average was used for analysis. When the difference between the two measurements was >0.5cm, the measurement was repeated, then the average of the repeated measurements was taken for analysis. To minimize incorporation bias, MUAC measurements were taken before weight and height measurements. Immediately after measuring MUAC, height and weight measurements were performed. The BMI Z score for all participants was calculated after the completion of the data collection, which avoids incorporation bias to the measurements.

To compute BMI Z score, height was measured barefoot with head in the Frankfort position to the nearest 0.1 cm and weight was measured barefoot with light cloth to the nearest 0.1 kg using a digital scale. To ensure measurement accuracy, the scale was checked for zero reading before each participant and calibrated regularly with an iron bar of 5 Kg. Weight and height for each participant were measured twice and the average was used for analysis.

To define overweight (including obesity), we used the World Health Organization BMI Z score reference. BMI Z score >+1SD is considered as overweight (including obesity) [19], BMI Z score is chosen as a reference test since a high BMI Z score can be an indicator of high body fatness. Even though BMI Z score does not measure body fat directly, it is correlated with direct measures of body fat [20, 21]. Furthermore, it is the most commonly used tool and the only available method in resource-limited settings like Ethiopia [6].

All anthropometric measurers had participated in a standardization exercise. The anthropometric measurers took repeated measurements of ten adolescents in two teams, one measurer each. Each measurer took two height, weight, and MUAC measurements for ten participants. We then compared the technical error of measurement for weight, height, and

MUAC with reference values [22]. All the technical errors of measurements were within the acceptable range.

## Statistical analysis

Data were entered using EpiData version 4.4.2.0 and exported to STATA version 15.1 for further processing and analysis. The data of participants with missing measurements either for MUAC, weight, or height were excluded from the analysis. Descriptive statistics including mean/median, standard deviation (SD), and percentages were applied to summarize the study participants characteristics. Frequency (percentages) was used to estimate the prevalence of overweight and obesity among adolescents. For continuous variables (MUAC, BMI Z score and age), normality was checked using Shapiro-Wilk normality test and visualized using Q-Q plots. We found that the data have a deviation from normality for MUAC, BMI Z score, and age (P-value <0.05).

Hence, we conducted a Spearman's rank correlation with 95% confidence interval to obtain insight into the strength of linear relationship between MUAC, BMI Z score, and age.

We computed the area under the ROC curve (discrimination) and a calibration plot (calibration) to evaluate the accuracy of MUAC to identify overweight/obesity among adolescents. The area under the ROC curve (AUC) determines the overall level of accuracy. An AUC of 0.5 indicating no predictive ability higher than random chance, whereas AUC of 1 indicates perfect diagnostic performance. The categories used to summarize the accuracy of AUC in ROC analysis were as follows: excellent (0.9–1), good (0.8–0.9), fair (0.7–0.8), poor (0.6–0.7) and fail (0.5–0.6) [23]. We also constructed a calibration curve of the predicted probability (using MUAC) in the x-axis against the true probability of overweight (BMI) in the y-axis. To examine the accuracy of MUAC between the sexes, we performed a sub-group analysis for males and females separately. The AUCs were adjusted for overfitting or over-optimism using a bootstrapping technique [24]. To this end, we draw 1000 random bootstrap samples with replacement from the dataset with complete data for MUAC and BMI Z score. The predictive performance after bootstrapping is considered as the performance that can be expected when MUAC is applied to future similar populations. The optimism coefficient was computed by subtracting the original performance measure from the AUC after bootstrapping ($AUC_{boot}$-$AUC_{origina}$). The optimal cut-off point was determined using the highest Youden index ($J = Sensitivity + Specificity—1$) [25].

The discriminatory ability and predictive value of MUAC cut-off points against BMI Z score $\geq$ +1 SD. MUAC compared to BMI Z score were assessed using sensitivity, specificity, positive predictive value, negative predictive value, positive and negative likelihood ratio with 95% confidence interval. Sensitivity is the proportion of true positive (adolescents classified as overweight/obese by MUAC and BMI Z score) in the total adolescents classified as overweight/obese by BMI Z score: TP/(TP+FN). Specificity is the proportion of true negative (adolescents classified as non-overweight/non-obese by MUAC and BMI Z score) in the total adolescents classified as non-overweight/non-obese obese by BMI Z score: TN/(TN+FP). Negative predictive value tells us how likely an adolescent is not overweight and obese if categorized by MUAC as non-overweight/non-obese: TN/(TN+FN). Positive predictive value tells us how likely an adolescent is to be overweight and obese if categorized by MUAC as overweight/obese: TP/(TP+FP). Negative likelihood ratio tells us how an adolescent is not overweight/obesity based on BMI z score is more likely to be categorized as non-overweight/non-obese by MUAC as compared to an adolescent with overweight/obesity based on BMI Z score: (1-sensitivity) / specificity. Positive likelihood ration tells us how much more likely the MUAC

categorized overweight/ obesity result is to occur in subjects with overweight/obesity compared to those without overweight and obesity: sensitivity / (1 –specificity).

This study is reported in accordance with the STARD (Standards for Reporting Diagnostic accuracy studies) 2015 statement [26], which included a 30-item checklist to give guidance for reporting (Table 1 in S1 Text).

## Participant consent and ethical approval

First, ethical clearance was obtained from the ethical review board of Addis Ababa University. Then a support letter was obtained from Addis Ababa University, School of Public Health, and submitted to Addis Ababa City Education Bureau. Permission was obtained from the education departments of sub-cities and the school principals of selected schools. Written informed consent was obtained from all adolescents aged greater than 18 years, whereas for those aged below 18 parental assent was obtained.

## Results

Out of 877 adolescents who were approached,851 has participated in the study. Twenty-six students were not included into the study due to the following reasons: twenty-one were absent on the scheduled days, five of them refused to remove their shoes and heavy clothes for anthropometric measurement (Fig 1).

A total of 851 adolescents, 456 males, and 395 females participated in this study. The mean and standard deviation of age, MUAC, and BMI Z score for the total participants were 16.7 (±1.1) years, 25.5 (±3.3) cm, and 0.44 (±1.2) respectively (Table 1).

### Prevalence of overweight and obesity

The overall prevalence of overweight among high school adolescents in Addis Ababa was 11.2% (95% CI; 9.2–13.5%), whereas the prevalence of obesity was 3.3% (95% CI; 2.3–4.7%) (Fig 2).

### Relationship between MUAC, BMI Z score, and age

We found that MUAC was strongly correlated with BMI Z score, $r = 0.81$ (95% CI; 0.79–0.84). However, MUAC was poorly correlated with adolescents' age, $r = 0.15$ (95% CI; 0.08–0.21).

### The ROC and calibration of MUAC to diagnose overweight among adolescents

Overall, the area under the ROC (AUC) of MUAC was 0.96 (95% CI; 0.94–0.97). The AUC after bootstrapping was 0.95 (95% CI; 0.94–0.96) with the average optimism of 0.007. The calibration graph shows that despite minimal underestimation at very low risk, the calibration was on average acceptable and the calibration test was not statistically significant (P-value = 0.06) (Figs 3 and 4).

The AUC for MUAC against our reference method (BMI Z score defined overweight) was excellent for males 0.96 (95% CI; 0.93–0.98) and females 0.96 (95% CI; 0.94–0.98). (Figs 5 and 6)

Based on the Youden index, the optimal MUAC cut-offs to identify overweight were 27.75 cm for males and 27.9 cm for females. This cutoff point gives high sensitivity and specificity for both males and females (sensitivity 94.1%, 90.3%; specificity 89.1%, 90.7% respectively). Moreover, MUAC can correctly identify the majority of adolescents with or without overweight (89.1% for males and 90.7% for females). (Tables 2 and 3).

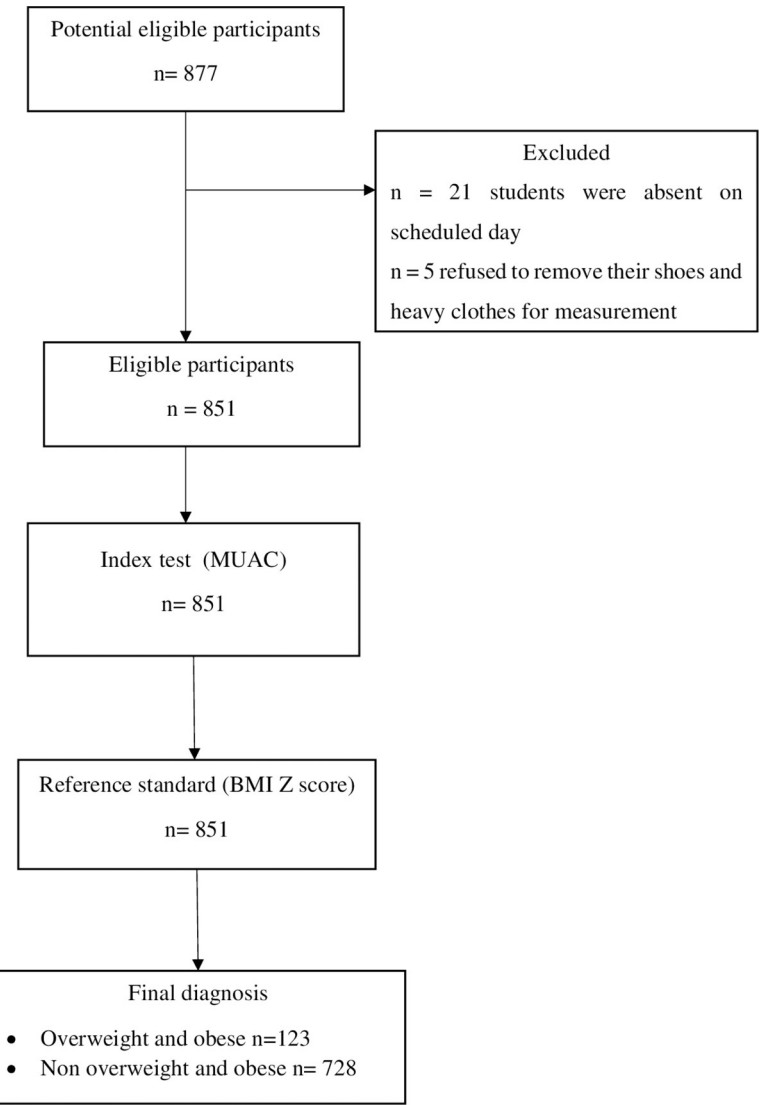

**Fig 1. The flow of participants through the study.**

**Table 1. Characteristics of study participants stratified by sex (n = 851).**

| Variables | Males (n = 456) | Females (n = 395) | Total (n = 851) |
|---|---|---|---|
| | Mean (±SD) | Mean (±SD) | Mean (±SD) |
| Age (years) | 16.8 ± 1.17 | 16.6 ±1.0 | 16.7 ± 1.1 |
| Height (cm) | 168.7 ± 6.8 | 157.0 ± 6.3 | 163.3 ± 8.8 |
| Weight (Kg) | 56.6 ±10.4 | 52.8 ± 10.3 | 54.9 ± 10.6 |
| BMI Z score (SD) | - 0.8 ± 1.2 | -0.05 ± 1.1 | -0.44 ± 1.2 |
| MUAC (cm) | 25.3 ± 3.2 | 25.7 ± 3.4 | 25.45 ± 3.33 |

BMI, body mass index; MUAC, mid-upper arm circumference.

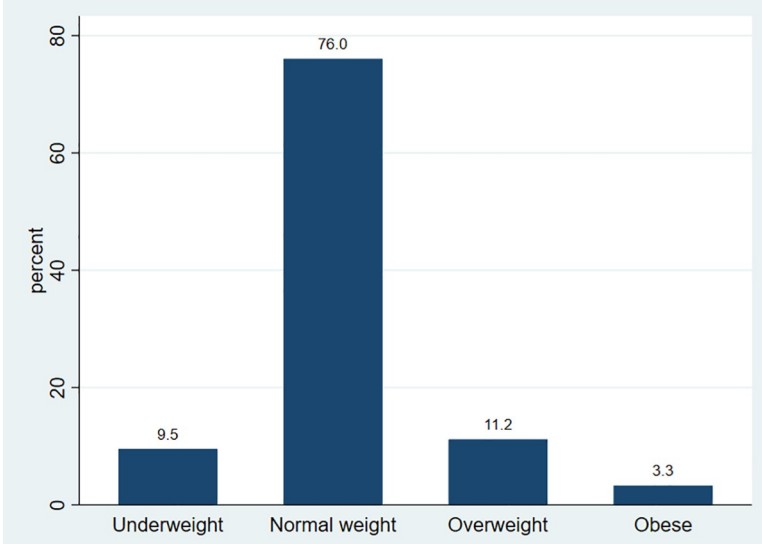

**Fig 2. Nutritional status of high school adolescents in Addis Ababa, Ethiopia, 2019.**

## Discussion

The present study showed that MUAC is an alternative measurement tool to identify overweight (including obesity) in adolescents aged 15–19 years. MUAC was strongly associated with BMI Z score for the total sample, suggesting that it can identify overweight in adolescents as accurate as BMI Z score. The AUC results (i.e., 0.96) showed MUAC has relatively equivalent diagnostic performance compared to BMI Z score in identifying adolescents with overweight/obesity.

This study found that MUAC has a high area under the AUC. Our study is supported by the findings of a recent study conducted among Chinese children aged 7–12 years reported an

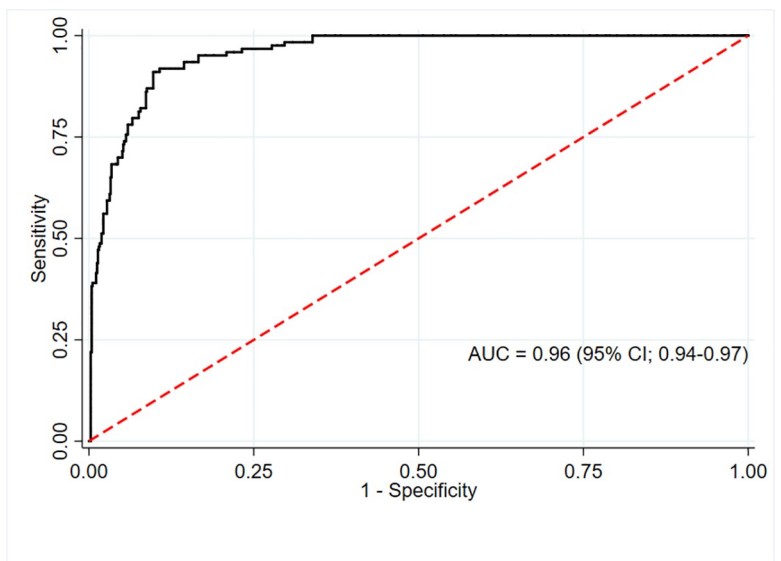

**Fig 3. ROC curve showing performance of MUAC to identify overweight / obesity in adolescents (n = 851).**

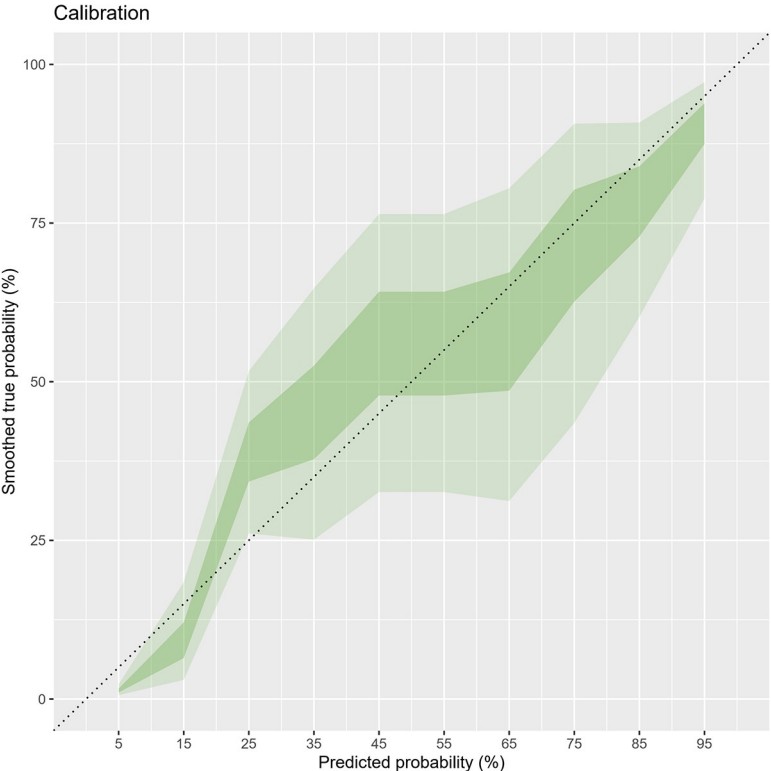

**Fig 4. Calibration of MUAC for identifying overweight/obesity adolescents (n = 851).**

AUC value range between 0.93 and 0.98 based on selected age and sex (MUAC vs. BMI Z score defined overweight/obesity) [13]. Likewise, a study done on Black South African children and adolescents aged 5–14 years showed MUAC has AUC values range between 0.90 and 0.97 as compared to BMI Z score to identify overweight [8]. Another study conducted on Indian children and adolescents aged 5–14 years showed MUAC had an AUC value range between 0.92 and 0.98 for identifying overweight [14].

Relevant studies that compared percent body fat with MUAC and BMI are scant. A study by Craig E. and his associates evaluated the performance of MUAC in comparison with BMI Z score %body fat measured by bioelectrical impendency, and found that MUAC accuracy was higher for BMI than for %body fatness [8]. However, due to lack of studies that compare MUAC and BMI with the reference standard, i.e. %body fat (total body water or multi-component methods), it is still inconclusive whether MUAC or BMI has a better accuracy [8, 27–30].

The present study found that, the optimal MUAC cut-off points to identify adolescent overweight are 27.75 and 27.9 cm for males and females, respectively. In previous studies, the proposed optimal MUAC cut-off to identify overweight/obesity range between 22.2 and 25.5 cm among study participants (age ranged between 7 and 15 years) [8, 13, 31]. In addition, the proposed cut off points to identify overweight among Turkish adolescents aged 15–17 years old range between 24.9 and 25.7 cm depending on age [29]. However, cut-off points determined by our study are higher than those reported by the previous studies. This might be due to an increase in MUAC with age; late adolescents (15–19 years) MUAC is expected to be higher than that of adolescents (10–17 years), this might result a higher cut-off point in late adolescents.

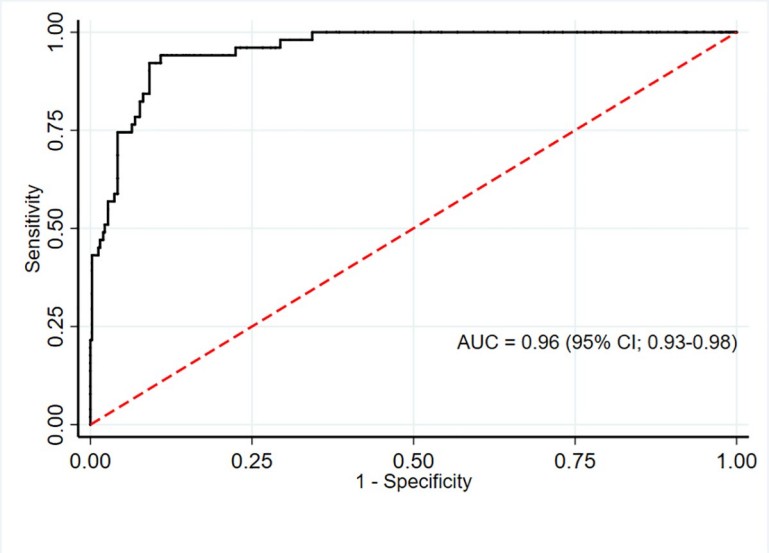

**Fig 5. ROC curve showing performance of MUAC to identify overweight/obesity in adolescent males (n = 456).**

The present study provides evidence that MUAC may also be used as an alternative tool to measure overweight/obesity in late adolescents aged 15–19 years. Color-coded MUAC tape, red for obese, amber for overweight, and green for normal weight may also be considered for non-numerate field workers to facilitate screening [27]. MUAC is weakly associated with the age of participants, this indicates adjustments may be necessary for the age of adolescents.

An ideal measure for detecting adolescent overweight and obesity should be reliable, inexpensive and easy to use [13]. While, evaluating MUAC as a measure of overweight and obesity has several key advantages: inexpensive, only a measuring tape is required, the measurements can be done easily in communities or schools, the interpretation can be easily understood by adolescents and families.

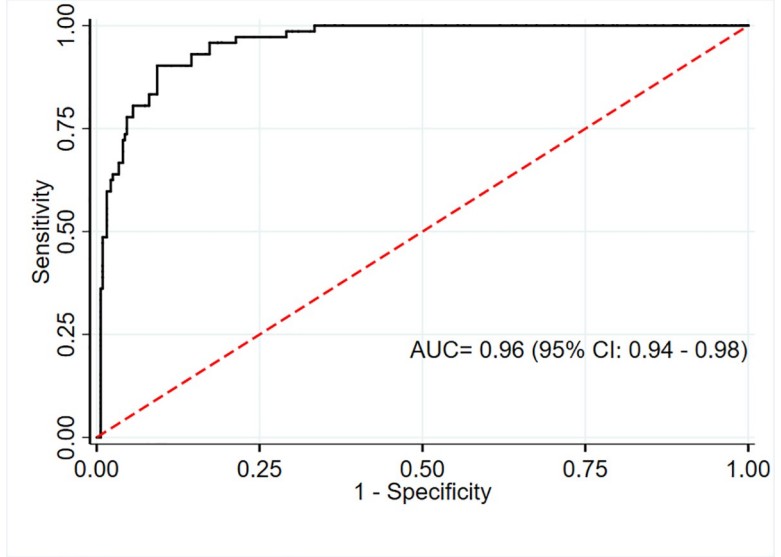

**Fig 6. ROC curve showing performance of MUAC to identify overweight/obesity in adolescent females (n = 395).**

**Table 2. Screening test result for BMI Z score defined overweight and obesity with MUAC among the adolescents (n = 851).**

| Overweight and obesity according to optimal MUAC cut-offs [1] | Overweight and obesity according to BMI Z score | | Total |
|---|---|---|---|
| | Yes | No | |
| Yes | 112 | 71 | 183 |
| No | 11 | 657 | 668 |
| Total | 123 | 728 | 851 |

[1]Optimal cutoff will be estimated using the Youden index from our data.

This study has its own strength and limitations. The strength of the present study is that we used a standardized measurement protocol and rigorous quality control measures to ensure high-quality data. The main limitation of this study is that we did not use the gold standard measures of percentage body fat, due to lack of equipment for the gold standard measures (total body water or multi-component methods) in our setting. BMI Z score is a commonly used method to identify adolescents with overweight and obesity. Although BMI Z score is correlated with percent body fat, it cannot distinguish between lean and fat mass [20]. Since we compare MUAC to BMI Z score, MUAC will have similar limitations to BMI Z Score. Moreover, adjustments may also be necessary for age given that the age of adolescents has been found to impact BMI Z score and body fat composition, but due to relatively small sample size, this study could not estimate age and sex specific MUAC cut-offs. Even though obesity is more important than overweight in regard to the risk of metabolic syndrome and adverse health outcomes, we were not able to determine cut-offs specifically for obesity due to the relatively small sample size.

## Conclusion

In conclusion, MUAC has relatively equivalent accuracy with BMI Z score to identify overweight /obesity among 15–19 years old adolescents. Hence, MUAC could be used as an alternative tool for surveillance and screening of overweight in adolescents aged 15–19 years in Ethiopia. We recommend future studies to evaluate the accuracy of MUAC compared to the reference standard indicators of adiposity (total body water or multi-component methods), with a nationally representative and adequate sample size for each sex and age group. Determining the age and sex specific cutoff points for obesity is also recommended. We further

**Table 3. Area under the receiver operating characteristics curve, sensitivities, specificities, positive predictive values, negative predictive values, positive likelihood ratio, negative likelihood ratio, correctly classified, Youden index, and optimal cut-off values of mid-upper-arm circumference in predicting overweight (n = 851).**

| Sex | Sensitivity (%) (95% CI) | Specificity (%) (95% CI) | PPV (%) (95% CI) | NPV (%) (95% CI) | LR+ (95% CI) | LR− (95% CI) | Correctly classified (%) | Youden index | Cut off point (cm) |
|---|---|---|---|---|---|---|---|---|---|
| Males | 94.1 (83.8–98.8) | 89.1 (85.7–92) | 52.2 (45–59.3) | 99.2 (97.6–99.7) | 8.7 (6.5–11.5) | 0.07 (0.0–0.2) | 89.7 | 0.83 | ≥ 27.75 |
| Females | 90.3 (81–96) | 90.7 (87–93.6) | 68.4 (60.4–75.4) | 97.7 (95.4–98.8) | 9.7 (6.9–13.8) | 0.11 (0.1–0.2) | 90.6 | 0.81 | ≥ 27.90 |
| Total | 91.1 (84.6–95.5) | 90.3 (87.9–92.3) | 61.7 (56.2–66.9) | 98.3 (97.1–99) | 9.3 (7.4–11.7) | 0.10 (0.1–0.2) | 90.4 | 0.81 | ≥ 27.95 |

CI, confidence interval; LR+, positive likelihood ratio; LR-, negative likelihood ratio, NPV, negative predictive value; PPV, positive predictive value.

suggest future studies to compare weather MUAC or BMI Z score is more accurate in comparison with the reference standard techniques of total body fat measures.

## Supporting information

**S1 Dataset. The minimal dataset of the study.**
(DTA)

**S1 Text. Standards for Reporting Diagnostic accuracy studies (STARD) checklist used in this study.**
(DOCX)

**S1 Table. Ability of MUAC to classify overweight and obesity among male adolescents, Addis Ababa, 2019 (n = 456).**
(DOCX)

**S2 Table. Ability of MUAC to classify overweight and obesity among female adolescents, Addis Ababa, 2019 (n = 395).**
(DOCX)

**S3 Table. Sensitivity, specificity, positive predictive value, negative predictive value, positive likelihood ratio, negative likelihood ratio, Youden index and optimal cut-off values of mid-upper-arm circumference in predicting overweight (including obesity) in adolescent males (n = 456).**
(DOCX)

**S4 Table. Sensitivity, specificity, positive predictive value, negative predictive value, positive likelihood ratio, negative likelihood ratio, Youden index and optimal cut-off values of mid-upper-arm circumference in predicting overweight (including obesity) in adolescent females (n = 395).**
(DOCX)

**S5 Table. Ability of MUAC to classify obesity among adolescent, Addis Ababa, 2019.**
(DOCX)

**S6 Table. Sensitivity, specificity, positive predictive value, negative predictive value, positive likelihood ratio, negative likelihood ratio, Youden index, and optimal cut-off values of mid-upper-arm circumference in predicting obesity (including obesity) (n = 851).**
(DOCX)

## Acknowledgments

We are very much thankful to all study participants for their willingness to participate in the study.

## Author Contributions

**Conceptualization:** Binyam Girma Sisay, Seifu Hagos Gebreyesus.

**Data curation:** Binyam Girma Sisay.

**Formal analysis:** Binyam Girma Sisay, Seifu Hagos Gebreyesus.

**Funding acquisition:** Binyam Girma Sisay.

**Investigation:** Binyam Girma Sisay.

**Methodology:** Binyam Girma Sisay, Demewoz Haile, Hamid Yimam Hassen, Seifu Hagos Gebreyesus.

**Project administration:** Binyam Girma Sisay, Seifu Hagos Gebreyesus.

**Supervision:** Binyam Girma Sisay, Demewoz Haile, Seifu Hagos Gebreyesus.

**Validation:** Hamid Yimam Hassen.

**Writing – original draft:** Binyam Girma Sisay, Demewoz Haile, Seifu Hagos Gebreyesus.

**Writing – review & editing:** Binyam Girma Sisay, Demewoz Haile, Hamid Yimam Hassen, Seifu Hagos Gebreyesus.

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
