## [Decision Letter · Decision Letter 0]

9 Mar 2020

PONE-D-19-36028

Performance of mid-upper arm

circumference as a screening tool for identifying adolescents with overweight and obesity

PLOS ONE

Dear Dr Gebreyesus,

Thank you for submitting your manuscript to PLOS ONE. After careful consideration, we feel that it has merit but does not fully meet PLOS ONE’s publication criteria as it currently stands. Therefore, we invite you to submit a revised version of the manuscript that addresses the points raised during the review process.

It is an interesting paper; however, major considerations reported by reviewers should be addressed. The main issues which I wish to hear from the authors are about the calibration of MUAC against BMI-for age and the possible influence of ethnicity in cutoff points of MUAC.

We would appreciate receiving your revised manuscript by April 6, 2020. To enhance the reproducibility of your results, we recommend that if applicable you deposit your laboratory protocols in protocols.io, where a protocol can be assigned its own identifier (DOI) such that it can be cited independently in the future. For instructions see: http://journals.plos.org/plosone/s/submission-guidelines#loc-laboratory-protocols

We look forward to receiving your revised manuscript.

Kind regards,

Joao Felipe Mota

Academic Editor

PLOS ONE

Journal Requirements:

"The funders had no role in study design, data collection and analysis, decision to publish, or preparation of the manuscript.".

a)    Please provide an amended Funding Statement that declares *all* the funding or sources of support received during this specific study (whether external or internal to your organization) as detailed online in our guide for authors at http://journals.plos.org/plosone/s/submit-now.  

b)    Please state what role the funders took in the study.  If any authors received a salary from any of your funders, please state which authors and which funder. If the funders had no role, please state: "The funders had no role in study design, data collection and analysis, decision to publish, or preparation of the manuscript."

6. Please upload a copy of Supporting Information (S1 Dataset) which you refer to in your text on page 23.

Additional Editor Comments (if provided):

It is an interesting paper; however, major considerations reported by reviewers should be addressed. The main issues which I wish to hear from the authors are about the calibration of MUAC against BMI-for age and the possible influence of ethnicity in cutoff points of MUAC.

Reviewers' comments:

Reviewer's Responses to Questions

**Comments to the Author**

1. Is the manuscript technically sound, and do the data support the conclusions?

Reviewer #1: Yes

Reviewer #2: Yes

Reviewer #3: Partly

2. Has the statistical analysis been performed appropriately and rigorously? 

Reviewer #1: Yes

Reviewer #2: Yes

Reviewer #3: Yes

3. Have the authors made all data underlying the findings in their manuscript fully available?

Reviewer #1: Yes

Reviewer #2: Yes

Reviewer #3: Yes

4. Is the manuscript presented in an intelligible fashion and written in standard English?

Reviewer #1: Yes

Reviewer #2: Yes

Reviewer #3: Yes

5. Review Comments to the Author

Reviewer #1: PONE-D-19-36028

The revised manuscript presented an interesting approach. The study was carefully conducted and it is well written. The discussion appropriately addresses the main results observed in the study. The conclusion answers the purpose of the study.

Please find below the minor amendments suggested:

- The acronym MUAC was used in line 22, then the authors should use it from this first mention (for example in lines 24, 26, 71, ...)

- Line 38. Please add 95% CI for AUAC.

- Line 138. It is necessary to offer more details about: If normality test was performed before the Spearman’s rank correlation?

- Line 184. Please add the meaning of the abbreviations BMI and MUAC in the footnote (Table 1).

- Line 245-251. It is important to discuss the findings by comparing those with studies that evaluated adolescents aged over 15 years. For example: Mazıcıoğlu MM, Hatipoğlu N, Oztürk A, Ciçek B, Ustünbaş HB, Kurtoğlu S. Waist circumference and mid-upper arm circumference in evaluation of obesity in children aged between 6 and 17 years. J Clin Res Pediatr Endocrinol. 2010; 2(4):144–150. doi:10.4274/jcrpe.v2i4.144

- Line 274 – The limitations of the present study could be minimized by presenting the results of previous studies. For example, Taylor at al. concluded “Categorisation of BMI according to both age and pubertal stage of development does not produce cutoffs that are superior to BMI cutoffs calculated on the basis of age alone at identifying children with high DXA-measured adiposity”.

Taylor, R., Falorni, A., Jones, I. et al. Identifying adolescents with high percentage body fat: a comparison of BMI cutoffs using age and stage of pubertal development compared with BMI cutoffs using age alone. Eur J Clin Nutr. 2003; 57:764–769. https://doi.org/10.1038/sj.ejcn.1601608.

Reviewer #2: The study is well performed, but misses some essential aspects that should be clarified before I can review the article more in detail. This article could be a contribution to public health. For further information see attachement.

Reviewer #3: This is a generally well presented manuscript which is written very clearly in the main. The design and analyses are well chosen, but I have serious reservations with the premise on which it is based which is that the BMI-for-age cut-off for overweight and obesity is an appropriate definition against which to calibrate an equivalent MUAC cut-off.

The study deals with the important issue of whether or not simpler alternative proxies (to the BMI-for-age) for high body fatness might be valid. The authors have derived a high MUAC cut-off which corresponds to BMI-for -age cut-offs equivalent to overweight and obesity and reached the conclusion that the MUAC is likely to provide acceptable agreement with high BMI-for-age. This is fine statistically as far as I can see, and there is a case that MUAC is likely to be more practical in LMICs. However, the problem is that the BMI-for-age cut-off to define overweight and obesity is actually very poor (as has been shown in a number of systematic reviews) and so calibrating a MUAC cut-off which corresponds to it (and suggesting that this MUAC cut off might then be used) might encourage use of this new MUAC cut-point which is likely to be equally flawed to BMI-for age.

I have a number of specific comments:

Abstract

line 38 characteristic singular

Introduction

line 47 change 'calorie' to energy (calorie is the unit of measurement not the variable);

line 67 adolescent (no s)

Methods

These are described well and generally sound apart from the flawed premise noted above, and it would be important to explain/justify the rationale for calibrating cut-offs against BMI for age defined overweight and obesity in general and also why calibrate against overweight as opposed to obesity (since obesity is more important, and has been related to comorbidities in a way which overweight has not).

It would have been useful to follow explicitly the guidance on studies of this kind which are available e.g. STARD.

Sampling and power are strong and unusually good for this type of study.

Line 127 delete 'Adolescents with'.

It would be useful to define/explain concepts and terms such as 'optimism' here. The other concepts and terms are much better known but it would also be useful to define these briefly in the text: sensitivity; specificity; positive and negative predictive values.

Results

Line 197 should be characteristic singular

Line 198 should be was not is (past tense).

Lines 205-207 notes excellent agreement but clarify that this is at the optimum cut-off point.

Have data been made available ?- I might have missed this.

Discussion

While the authors acknowledge the weakness of not calibrating MUAC against a reference method (gold standard) on lines 269-272 I don't think they go nearly far enough here and the implications of their findings are potentially harmful. Since BMI-for age performs poorly as a proxy for obesity (obesity is excessive fatness and BMI for age has only low-moderate sensitivity but high specificity according to multiple systematic reviews) calibrating a MUAC cutpoint which is equivalent to a BMI for age cutpoint would simply replace one poor proxy by another. For that reason I think that this is a fundamental conceptual flaw in this study event though it is otherwise sound methodologically.

I also note that alternative methods of measuring body fatness more directly (DEXA is mentioned in the Discussion) are not reference methods, e.g. see Wells and Fewtrell Arch Dis Child 2006. The only reference or gold standard lab methods are multi-component and the only field reference method is total body water.

Line 245 adolescent singular.

6. PLOS authors have the option to publish the peer review history of their article (what does this mean?). If published, this will include your full peer review and any attached files.

Reviewer #1: No

Reviewer #2: Yes: H Talma, MD

Reviewer #3: No

---

## [Author Response · Author response to Decision Letter 0]

30 Mar 2020

Reviewer 1: We have incorporated all suggestions in to my revision. They were very helpful. Thank you 

Reviewer 2: we have explained thoroughly about racial identity of Addis Ababa in the response to reviewer word file. That was very helpful. Thank you

Reviewer 3: we have incorporated all suggestions in to my revision. They were very helpful. Thank you

---

## [Decision Letter · Decision Letter 1]

6 May 2020

PONE-D-19-36028R1

Performance of mid-upper arm circumference as a screening tool for identifying adolescents with overweight and obesity

PLOS ONE

Dear Dr Gebreyesus,

Thank you for submitting your manuscript to PLOS ONE. After careful consideration, we feel that it has merit but does not fully meet PLOS ONE’s publication criteria as it currently stands. Therefore, we invite you to submit a revised version of the manuscript that addresses the points raised during the review process.

As you can see the reviewers see real merit in your work, but also have concerns that need to be addressed within the manuscript before we can accept for publication. The reviewer #2 has attached their comments. Please consider an English language edit before to resubmit.

We would appreciate receiving your revised manuscript by June 5. To enhance the reproducibility of your results, we recommend that if applicable you deposit your laboratory protocols in protocols.io, where a protocol can be assigned its own identifier (DOI) such that it can be cited independently in the future. For instructions see: http://journals.plos.org/plosone/s/submission-guidelines#loc-laboratory-protocols

We look forward to receiving your revised manuscript.

Kind regards,

Joao Felipe Mota

Academic Editor

PLOS ONE

Additional Editor Comments (if provided):

As you can see the reviewers see real merit in your work, but also have concerns that need to be addressed within the manuscript before we can accept for publication. The reviewer #2 has attached their comments. Please consider an English language edit before to resubmit.

Reviewers' comments:

Reviewer's Responses to Questions

**Comments to the Author**

1. If the authors have adequately addressed your comments raised in a previous round of review and you feel that this manuscript is now acceptable for publication, you may indicate that here to bypass the “Comments to the Author” section, enter your conflict of interest statement in the “Confidential to Editor” section, and submit your "Accept" recommendation.

Reviewer #1: All comments have been addressed

Reviewer #2: (No Response)

Reviewer #3: (No Response)

2. Is the manuscript technically sound, and do the data support the conclusions?

Reviewer #1: Yes

Reviewer #2: Yes

Reviewer #3: Yes

3. Has the statistical analysis been performed appropriately and rigorously? 

Reviewer #1: Yes

Reviewer #2: Yes

Reviewer #3: Yes

4. Have the authors made all data underlying the findings in their manuscript fully available?

Reviewer #1: Yes

Reviewer #2: Yes

Reviewer #3: Yes

5. Is the manuscript presented in an intelligible fashion and written in standard English?

Reviewer #1: Yes

Reviewer #2: Yes

Reviewer #3: Yes

6. Review Comments to the Author

Reviewer #1: The authors accepted the reviewer’s suggestions for improving the manuscript. I highlight the following aspects as the most relevant:

They added more description about the normality test and its result.

Accepted the suggestion of Reviewer 3: sensitivity, specificity, positive and negative predictive values were defined. The description of how computed the optimism coefficient also was add.The main limitation of the study: not calibrating MUAC against a reference method (gold standard) was properly explored in the discussion.

Besides, the manuscript has been revised according to STARD 2015 guideline. The authors have attached STARD 2015 checklist as supplementary file.

Reviewer #2: The impact of this paper could have been greater when it was age related/specific, one cutoff for the whole range from 15-19 years (sex-specific) is not specific eneough and not comparable with other studies.

Reviewer #3: The authors have partially addressed the major concern that I had at the previous review stage, that use of BMI Z score as the reference method was limited because that is a poor indicator of excess fatness. The authors now acknowledge this important point in the revised discussion, but the language used in places in the Discussion still tends to ignore this difficulty and in doing so overstates the accuracy of MUAC. In particular Discussion lines 252 and 255 overstate accuracy- accuracy of MUAC cannot be 'excellent' because it is being judged against a reference method (BMI Z score) which itself is not excellent.In summary, the authors should use more careful, appropriate, language in lines 252 and 256, or should omit these exaggerated claims.

Two minor points should be addressed:

1. The manuscript is generally well written and clear, but there are a few minor grammatical errors and I think it would benefit from an English language edit;

2. Line 288- the gold standard is not 'doubly labelled water' and so this phrase should be replaced with ' total body water or multi-component methods (which measure body density, total body water, and total body mineral'.

7. PLOS authors have the option to publish the peer review history of their article (what does this mean?). If published, this will include your full peer review and any attached files.

Reviewer #1: No

Reviewer #2: Yes: Henk Talma

Reviewer #3: No

---

## [Author Response · Author response to Decision Letter 1]

2 Jun 2020

Dear reviewers we have addressed all your comments in the response to reviewer word file.

---

## [Editor Report · Decision Letter 2]

9 Jun 2020

Performance of mid-upper arm circumference as a screening tool for identifying adolescents with overweight and obesity

PONE-D-19-36028R2

Dear Dr. Gebreyesus,

We’re pleased to inform you that your manuscript has been judged scientifically suitable for publication and will be formally accepted for publication once it meets all outstanding technical requirements.

Kind regards,

Joao Felipe Mota

Academic Editor

PLOS ONE
---

## [Editor Report · Acceptance letter]

11 Jun 2020

PONE-D-19-36028R2 

Performance of mid-upper arm circumference as a screening tool for identifying adolescents with overweight and obesity 

Dear Dr. Gebreyesus:

I'm pleased to inform you that your manuscript has been deemed suitable for publication in PLOS ONE. Congratulations! Your manuscript is now with our production department. 

Kind regards, 

on behalf of

Dr. Joao Felipe Mota 

Academic Editor

PLOS ONE